# Correlation of RECIST, Computed Tomography Morphological Response, and Pathological Regression in Hepatic Metastasis Secondary to Colorectal Cancer: The AVAMET Study

**DOI:** 10.3390/cancers12082259

**Published:** 2020-08-12

**Authors:** Ruth Vera, María Luisa Gómez, Juan Ramón Ayuso, Joan Figueras, Pilar García-Alfonso, Virginia Martínez, Adelaida Lacasta, Ana Ruiz-Casado, María José Safont, Jorge Aparicio, Juan Manuel Campos, Juan Carlos Cámara, Marta Martín-Richard, Clara Montagut, Carles Pericay, Jose María Vieitez, Esther Falcó, Mónica Jorge, Miguel Marín, Mercedes Salgado, Antonio Viúdez

**Affiliations:** 1Medical Oncology Department, Complejo Hospitalario de Navarra, Instituto de investigaciones Sanitarias de Navarra (IdISNA), 31008 Pamplona, Spain; antonio.viudez.berral@navarra.es; 2Pathology Department, Complejo Hospitalario de Navarra, 31008 Pamplona, Spain; ml.gomez.dorronsoro@navarra.es; 3Radiology Department, Hospital Clínic de Barcelona, 08036 Barcelona, Spain; JRAYUSO@clinic.cat; 4General and digestive surgery Department, Hospital Universitario Josep Trueta, 17007 Girona, Spain; jfigueras.net@icloud.com; 5Medical Oncology Department, Hospital General Universitario Gregorio Marañón, 28007 Madrid, Spain; pgarcaalfonso@gmail.com; 6Medical Oncology Department, Hospital Universitario La Paz, 28046 Madrid, Spain; virgimarin9@hotmail.com; 7Medical Oncology Department, Hospital Universitario Donostia, 20014 San Sebastian, Spain; ADELAIDA.LACASTAMUNOA@osakidetza.eus; 8Medical Oncology Department, Hospital Universitario Puerta de Hierro, 28222 Majadahonda, Spain; aruiz.hflr@salud.madrid.org; 9Medical Oncology Department, Hospital General Universitario de Valencia, 46014 Valencia, Spain; mjsafont@yahoo.es; 10Medical Oncology Department, Hospital Universitario y Politécnico La Fe, 46026 Valencia, Spain; japariciou@seom.org; 11Medical Oncology Department, Hospital Arnau de Vilanova, 46015 Valencia, Spain; jmcamposcervera@yahoo.es; 12Medical Oncology Department, Hospital Universitario Fundación Alcorcón, 28922 Madrid, Spain; jccamara@fhalcorcon.es; 13Medical Oncology Department, Hospital la Santa Creu i Sant Pau, 08041 Barcelona, Spain; mmartinri@santpau.cat; 14Medical Oncology Department, Hospital de Mar, 08003 Barcelona, Spain; CMontagut@parcdesalutmar.cat; 15Medical Oncology Department, C.S. Parc Taulí, 08208 Sabadell, Spain; cpericay@gmail.com; 16Medical Oncology Department, Hospital Universitario Central de Asturias, 33011 Oviedo, Spain; josemariavieitez@yahoo.es; 17Medical Oncology Department, Hospital Universitario Son Llàtzer, 07198 Palma de Mallorca, Spain; efalco@hsll.es; 18Medical Oncology Department, Hospital Xeral Cíes, 36204 Vigo, Spain; Monica.Jorge.Fernandez@sergas.es; 19Medical Oncology Department, Hospital Clínico Universitario de la Arrixaca, 30120 Murcia, Spain; miguelmarin75@hotmail.com; 20Medical Oncology Department, Complejo Hospitalario de Ourense, 32005 Ourense, Spain; Mercedes.Salgado.Fernandez@sergas.es

**Keywords:** metastatic colorectal cancer, neoadjuvant chemotherapy, bevacizumab, antiangiogenics, computed tomography-based morphological criteria

## Abstract

*Background*: The prospective phase IV AVAMET study was undertaken to correlate response evaluation criteria in solid tumors (RECIST)-defined response rates with computed tomography-based morphological criteria (CTMC) and pathological response after liver resection of colorectal cancer metastases. *Methods*: Eligible patients were aged ≥18 years, with Eastern Cooperative Oncology Group (ECOG) performance status 0/1 and histologically-confirmed colon or rectal adenocarcinoma with measurable liver metastases. Preoperative treatment was bevacizumab (7.5 mg on day 1) + XELOX (oxaliplatin 130 mg/m^2^, capecitabine 1000 mg/m^2^ bid on days 1–14 q3w). After three cycles, response was evaluated by a multidisciplinary team. Patients who were progression-free and metastasectomy candidates received one cycle of XELOX before undergoing surgery 3–5 weeks later, followed by four cycles of bevacizumab + XELOX. *Results*: A total of 83 patients entered the study; 68 were eligible for RECIST, 67 for CTMC, and 51 for pathological response evaluation. Of these patients, 49% had a complete or partial RECIST response, 91% had an optimal or incomplete CTMC response, and 81% had a complete or major pathological response. CTMC response predicted 37 of 41 pathological responses versus 23 of 41 responses predicted using RECIST (*p* = 0.008). Kappa coefficients indicated a lack of correlation between the results of RECIST and morphological responses and between morphological and pathological response rates. Conclusion: CTMC may represent a better marker of pathological response to bevacizumab + XELOX than RECIST in patients with potentially-resectable CRC liver metastases.

## 1. Introduction

Liver metastasis is common in patients with colorectal cancer (CRC)—25% of patients have liver metastases at diagnosis and metastases occur later during the disease course in a further 25–35% [1]. Resection of liver metastases can be effective in prolonging survival in suitable patients [2,3] and treatment approaches leading to the conversion of initially-unresectable into resectable metastases can improve outcomes in patients with metastatic CRC (mCRC) [4,5]. Consequently, treatment guidelines currently recommend tumor shrinkage followed by metastasectomy for patients with technically-resectable metastases where the prognosis is unclear or probably unfavorable [6]. Bevacizumab-based treatment regimens have been shown to be effective in downstaging liver metastases in patients with mCRC, resulting in a proportion of patients becoming suitable for surgical resection. In the OLIVIA study, bevacizumab in combination with 5-fluorouracil/folinic acid plus oxaliplatin and irinotecan (FOLFOXIRI) or modified 5-fluorouracil/folinic acid plus oxaliplatin (mFOLFOX) resulted in resection rates of 61% and 49%, respectively, in patients with initially-unresectable liver metastases from CRC [7].

Accurate assessment of response to treatment is essential in patients with liver-only metastases who may be candidates for potentially-curable surgery. Tumor response rates are traditionally calculated using size-based radiological criteria, such as response evaluation criteria in solid tumors (RECIST) [8]. RECIST may not adequately determine response to bevacizumab, as the cytostatic action of bevacizumab may have little impact on tumor size [9,10]. In their phase III study, Saltz and colleagues reported that the addition of bevacizumab to oxaliplatin-based chemotherapy significantly improved progression-free survival (PFS) without affecting RECIST response rates [11]. Others have reported morphological changes independent of reductions in tumor size but correlating with pathological responses following treatment with bevacizumab in patients with CRC liver metastases [9].

The prospective phase IV AVAMET study was undertaken to correlate pathological response after resection of liver with RECIST-defined response with morphological response using computed tomography-based morphological criteria (CTMC) in patients with CRC and resectable hepatic metastases undergoing perioperative treatment with bevacizumab combined with capecitabine plus oxaliplatin (XELOX). Secondary objectives were to evaluate safety and long-term survival after perioperative chemotherapy as well as to assess outcomes according to *KRAS*-mutation status in this patient population.

## 2. Patients and Methods

This study was performed at 22 institutions in Spain. Eligible patients were aged ≥18 years, with Eastern Cooperative Oncology Group (ECOG) performance status of 0/1 and histologically-confirmed colon or rectal adenocarcinoma with evidence of synchronous or metachronous measurable liver metastases (RECIST version 1.1). The primary tumor could have been resected before inclusion in the study or subsequently resected if the patient was not symptomatic. Patients were required to have resectable liver metastases, i.e., ≤4 metastases, <10 cm in size, with surgery theoretically resulting in R0 resection and residual hepatic volume of ≥30%. Patients could have bilateral metastases provided there were ≤4 metastases and these were <10 cm in size. Other inclusion criteria were: tumor sample available for determination of KRAS mutation status; no prior therapy for mCRC; and adequate bone marrow, kidney, and liver function.

Patients were excluded if they had unresectable liver metastases, prior systemic therapy for metastatic disease, evidence of extrahepatic metastatic disease, or neoadjuvant chemotherapy or radiotherapy in the 6 months before study entry. Invasion of the vena cava with invasion of ≥2 hepatic veins, both portal veins, future liver remnant <30%, or portal embolization before hepatectomy were further exclusion criteria. Patients with hypertension that was not adequately controlled, peripheral neuropathy grade ≥1, significant vascular disease, history of hypertensive crisis, myocardial infarction or haemoptysis were also excluded, as were patients with major surgery in the four weeks before initiation of study treatment.

During the presurgical phase, patients received bevacizumab 7.5 mg on day 1, oxaliplatin 130 mg/m^2^ on day 1, and capecitabine 1000 mg/m^2^ twice daily on days 1–14 every 3 weeks (one cycle). After three cycles, patients were evaluated for response to treatment using RECIST (version 1.1) by a multidisciplinary team consisting of an oncologist, radiologist, and surgeon. Patients who were progression-free and were candidates for surgical resection received one cycle of XELOX, and 3–5 weeks later they underwent surgery. Postoperative chemotherapy consisted of four cycles of bevacizumab plus XELOX.

Patients with evidence of disease progression before or during surgery, and therefore not candidates for surgical resection, were withdrawn from active treatment, receiving further treatment according to the investigator’s preference. These patients were followed for evaluation of overall survival (OS).

This study was performed in accordance with the Declaration of Helsinki. Ethics approval was provided by the Comité Etico de Investigación Clínica de Navarra; all patients provided written, informed consent.

### 2.1. Outcomes

The primary endpoint of the study was correlation of pathological response after resection of liver metastases with RECIST-defined response and morphological response using CTMC. Secondary endpoints were relapse-free survival (RFS) in patients undergoing resection of metastases, OS, and safety. The impact of KRAS mutation on the primary objective was also examined.

Radiological response was evaluated centrally at the Diagnostic Radiology Service of Hospital Clínic i Provincial de Barcelona, Spain, using measurements taken before and after chemotherapy.

Tumor morphology was assessed using computed tomography and characterised according to previously-described criteria. Based on these measurements, metastases were assigned to one of three groups: group 1 if the tumor was homogeneous and hypoattenuating, with sharp tumor–liver interface, and no (or completely resolved) peripheral rim of enhancement; group 2 for tumors with mixed attenuation, variable tumor–liver interfaces, and partially resolved (in initially present) peripheral rim of enhancement; and group 3 for tumors with heterogeneous overall attenuation, poorly defined tumor–liver interfaces, and the possible presence of a peripheral rim of enhancement. Response was defined as optimal if the metastasis changed from Group 3 or 2 to Group 1, incomplete in the case of changes from Group 3 to Group 2, or null if the metastasis group was unchanged or increased.

The extent of residual carcinoma after resection of liver metastases was assessed semi-quantitatively as a percentage relative to the total tumor surface area. Pathological responses were scored as minor if ≥50% of tumor cells remained, major if 1–49% of tumor cells remained, or complete if no residual tumor cells were detected.

In patients with resected metastases, RFS was defined as the time from intervention to the relapse of metastases. Patients who were alive and progression-free were censored on the date of their last documented evaluation.

OS, which was determined in all patients, was defined as the time from the start of treatment to the time of death due to any cause. Patients who were still alive at the time of the OS analysis were censored on their last contact date.

### 2.2. KRAS Mutation Status Correlation

DNA was extracted from 3–5 µm hematoxylin–eosin-stained cuts of tumor tissue obtained from filed paraffin blocks. DNA extraction was carried out using the QIAamp DNA FFPE Tissue Kit (Qiagen, Germantown, MD, USA), with quantification of DNA concentration and quality carried out using the Nanodrop equipment (Thermo Scientific, Waltham, MA, USA).

Mutation detection was based on quantitative PCR using the mutation-specific amplification refractory mutation system and the Scorpions^®^ detection techniques. The Qiagen KRAS RGQ PCR kit, validated for diagnostics, was used for this purpose. The 7300 Real-Time PCR system (Life Technologies, Thermo Scientific, USA) was used in a first time- step; RotorGene (Qiagen, USA) was the second time-step. This kit allows testing of seven mutations in exon 2 of the KRAS oncogene, six in codon 12 (c.34GGT>TGT, p.Gly12Cys; c.34GGT>CGT, Gly12Arg; c.34GGT>AGT, p.Gly12Ser; c.35GGT>GCT, p.Gly12Ala; c.35GGT>GAT, p.Gly12Asp; c.35GGT>GTT, p.Gly12Val), and one in codon 13 (c.38GGC>GAC, p. Gly13Asp).

### 2.3. Statistical Analysis

We assumed a major or complete pathological response rate of 57% and a minor pathological response rate of 42%, with 82% of the minor pathological responses corresponding to incomplete or null morphological response [9]. Further assuming an alpha error of 0.05 and a beta error of 0.2 (80% power), 74 evaluable patients would be needed to result in 31 patients with a minor pathological response. Accounting for a 10% attrition rate, a total of 83 patients would need to be entered into the study.

Cohen’s kappa coefficients (K) were calculated for correlation of RECIST (yes (complete response/partial response) and no (stable/progressive disease)), morphological (yes (optimal/incomplete) and no (null)) and pathological (yes (complete/major) and no (minor)) response rates. Coefficients were classified as: slight (K values 0.00–0.20); fair (K 0.21–0.40; moderate (K 0.41–0.60; good (K 0.61–0.80), and almost perfect (K 0.81–1.00). Two-sided *p*-values were calculated for the test of the null hypothesis (i.e., that K = 0).

Categorical variables were described in absolute numbers and percentages and compared using the chi-squared test. The log-rank test was used to evaluate means and medians of variables relating to survival.

## 3. Results

### 3.1. Patients

A total of 83 patients were entered into the study (Figure 1). Patient characteristics are summarised in Table 1. Of the 83 patients in the study and included in the intent-to-treat population, 68 were eligible for RECIST assessment, 67 for morphological response assessment, and 51 for pathological response evaluation.

A total of 59 patients underwent surgery after receiving treatment according to the trial design; 51 of these patients (86%) had resected liver metastases. Four patients for whom a metastasectomy was initially proposed received ablative therapies.

Resection of liver metastasis was attempted in 59 patients (76% of the safety population) and performed in 51 patients (65%).

### 3.2. Response to Treatment

Responses for evaluable patients are summarised in Table 2: 49% (95% confidence interval (CI), 37–62%) of patients had a complete or partial response according to RECIST, 91% of patients (95% CI, 81–96%) had a CTMC response of optimal or incomplete, and complete and major pathological responses were recorded in 24% (95% CI, 13–37%) and 57% (95% CI, 42–71%) of patients, respectively.

### 3.3. Correlation of RECIST and Morphological and Pathological Response Data

RECIST, morphological, and pathological response data were available for the 51 patients who underwent resection of metastases; correlation of these data are shown in Figure 2. Morphological response (optimal/incomplete) was more sensitive than RECIST (complete/partial response) for predicting pathological responses (complete/major)—morphological response predicted 37 of 41 pathological responses compared with 23 of 41 responses predicted using RECIST (*p* = 0.008).

Cohen’s kappa coefficients were as follows: K = 0.0498 (95% CI, −0.0803–0.1800; *p* = 0.458) for the correlation between RECIST and morphological response rates; K = 0.1665 (95% CI, −0.0520–0.3820; *p* = 0.1388) for the correlation between RECIST and pathological response rates; and K = −0.1262 (95% CI, −0.2211 to −0.0313; *p* = 0.3035) for the correlation between pathological and morphological response rates. These values indicate no agreement between any of the methods used to assess response in these patients. Similarly, no agreement was observed between any of the methods used to assess response in these patients according to the methodology described by Chun and colleagues [9].

### 3.4. Survival

After a median follow-up of 30.5 months (95% CI, 10.6–48.7 months) in the 51 patients who underwent resection of metastases, relapse of disease occurred in 23 patients. Mean RFS was 25.9 months (95% CI, 21–31 months). Median RFS was not reached.

At the time of the analysis, 24 patients had died, and 59 patients were censored. Median OS was not reached in the overall population (95% CI, 35.1–not evaluable; Figure 3). The 1-year survival rate was 92% (95% CI 83–96%); at 4 years, 59% (95% CI, 42–72%) of patients were alive. Median OS was also not reached in the cohort of patients who underwent surgery. Among the 51 patients who underwent liver analysis, 12 had died at the time of the analysis. One-year and 4-year survival rates were 98% and 66%, respectively.

### 3.5. KRAS Mutation Status Analysis

Patient characteristics according to KRAS mutation status are shown in Table 1. Characteristics were largely similar in patients with KRAS mutant and wild-type tumors (all *p* > 0.05). Radiological response rates according to KRAS status are shown in Table 2. There were no significant differences in radiological, morphological, or pathological response rates between patients with mutant or wild-type KRAS status, although there was a numerical difference in complete pathological response rates in favor of patients with KRAS wild-type tumors (31% vs. 14%; *p* = 0.347). Similar proportions of patients in both groups underwent surgery (mutant KRAS, 75%; wild-type KRAS, 70%).

### 3.6. Safety

Five of the 83 patients included in the study were considered screening failures and the safety population therefore consisted of 78 patients. Reasons for premature withdrawal are shown in Figure 1.

In the presurgical phase, 78, 74, and 71 patients completed bevacizumab cycles 1, 2, and 3, respectively. Cycles were delayed in 21 (26.9%) patients because of non-hematological toxicity (*n* = 11), hematological toxicity (*n* = 3), reasons that were not due to treatment (*n* = 4), or other reasons (*n* = 3). The dose of bevacizumab was reduced in 8 patients (10.2%) primarily because of a variation in body weight. A total of 78, 74, 71, and 58 patients received 1, 2, 3, and 4 cycles of oxaliplatin, respectively. Cycles were delayed in 31 (39.7%) patients because of non-hematological toxicity (*n* = 17), hematological toxicity (*n* = 6), reasons that were not due to treatment (*n* = 3), or other reasons (*n* = 5). The dose of oxaliplatin was reduced in 20 patients (25%). Capecitabine was administered to 78 patients; 21 (27%) had dose delays, 19 (24%) had dose reductions, and 14 (18%) had dose interruptions.

Six patients (10%) suffered grade III or V Clavien–Dindo complications after surgery—two patients had a liver abscess, one patient had a pelvic abscess, one had intra-abdominal collection, one had evisceration, and one had a jejunal perforation. There were no postoperative deaths.

In the post-surgical treatment phase, 42, 41, 37, and 31 patients received 1, 2, 3, and 4 cycles of bevacizumab, respectively, and 42, 40, 35, and 24 patients received 1, 2, 3, and 4 cycles of oxaliplatin, respectively. A total of 43 patients received capecitabine in the post-surgical phase.

All 78 patients in the safety population reported at least one adverse event (AE) and 43 patients (55%) had at least one grade 3/4 AE (Table 3). Serious AEs (SAEs) were recorded in 34 patients (44%), the most common of which were diarrhoea (*n* = 7; 9%) and pulmonary embolism (*n* = 4; 5%). Four patients had an AE that resulted in death—one had septic shock deemed to have a remote chance of relationship to treatment agents; one had palmar-plantar erythrodysaesthesia that was probably related to capecitabine; one had paralytic ileus possibly related to oxaliplatin; and one had a hepatic abscess that was deemed unrelated to treatment agents.

Adverse events (AEs), serious adverse events (SAEs), and grade 3/4 AEs potentially related to bevacizumab occurred in 60 (77%), 12 (15%), and 20 patients (26%), respectively. AEs, SAEs, and grade 3/4 AEs potentially related to oxaliplatin occurred in 73 (94%), 13 (17%), and 28 patients (36%), respectively. AEs, SAEs, and grade 3/4 AEs potentially related to oxaliplatin occurred in 74 (95%), 20 (26%), and 27 patients (35%), respectively. AEs of special interest with bevacizumab are shown in Appendix A.

## 4. Discussion

For patients with mCRC and liver-limited metastases, curation may be an attainable goal if resection of the metastases can be achieved. In the AVAMET study, patients with mCRC and resectable liver-only metastases received preoperative bevacizumab plus XELOX, followed by surgery and postoperative bevacizumab plus XELOX, with the aim of enabling R0 resection and prolonging survival. As patients had four or fewer metastases of <10 cm in size, surgery could theoretically result in R0 resection and residual hepatic volume of ≥30%.

Resection of liver metastasis was attempted in 59 patients (76% of the safety population) and performed in 51 patients (65%). This percentage only includes patients who followed the study protocol, undergoing resection after receiving treatment according to the trial design. Some patients with protocol violations underwent surgery but were not included in the analysis of resection rates; inclusion of these patients would have resulted in a higher rate similar to rates observed in other studies [12,13,14].

Among patients whose response could be assessed using RECIST, morphological, and pathological assessments, CTMC appeared to be more accurate than RECIST at predicting complete and major pathological responses. Notably, 17 patients with a RECIST response of stable disease had a complete (*n* = 5) or major (*n* = 12) pathological response and 27 patients with RECIST-defined stable disease had either an optimal (*n* = 10) or incomplete (*n* = 17) morphological response. These data support our earlier finding that RECIST may not be a good indicator of pathological response in this patient population [15]. Although CTMC appeared to be more specific for predicting a complete/major pathological response than RECIST, kappa coefficients indicated a lack of correlation between the results of RECIST and morphological responses and between morphological and pathological response rates. CTMC are based in three issues—heterogeneity and attenuation, tumor–liver interface, and peripheral rim enhancement. It is possible that one of these issues, in an independent way, was more sensitive in predicting histological regression. As suggested by others [10], the combined use of RECIST and morphological criteria, plus pathological response when available, is mandatory for optimal evaluation in this setting.

Analysis of survival was an exploratory endpoint of this study. At the time of the analysis, after almost three years of follow up, median RFS had not been reached in patients who underwent resection. Median OS had not been reached in the overall cohort of patients nor in those who underwent resection; 59% of patients who underwent resection were alive after four years. These data suggest that perioperative bevacizumab plus XELOX is a valid treatment option for patients with resectable colorectal liver metastases.

The safety of bevacizumab plus XELOX in this patient population was in line with previous reports and no new safety signals were observed [11,16,17]. Surgical complications affected 12% of patients undergoing resection of metastases. This is similar to complication rates observed in other studies [18].

Some limitations of the present study should be considered, including the number of patients who withdrew from the study. In addition, the change of surgical approach in a number of patients with response, from resection to ablative therapies, affected the resection rate. It is possible that treatment reduced the size of the metastases, making them amenable to ablative techniques rather than resection, although no information is available to confirm this.

## 5. Conclusions

Results of the AVAMET study suggest that CTMC may represent a better marker than RECIST of pathological response to bevacizumab plus XELOX in patients with potentially-resectable CRC liver metastases. Resection of liver metastases in patients with mCRC after preoperative bevacizumab plus XELOX was associated with excellent RFS and OS, indicating that, where feasible, this approach provides suitable patients with liver-only metastases an opportunity for improved outcomes.

## Figures and Tables

**Figure 1 cancers-12-02259-f001:**
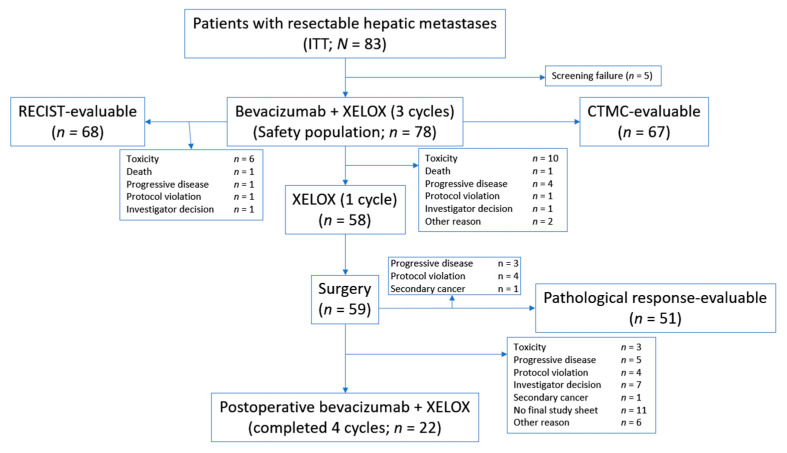
Patient flow. CTMC, computed tomography-based morphological criteria; ITT, intent to treat; RECIST, response evaluation criteria in solid tumors; XELOX, capecitabine plus oxaliplatin.

**Figure 2 cancers-12-02259-f002:**
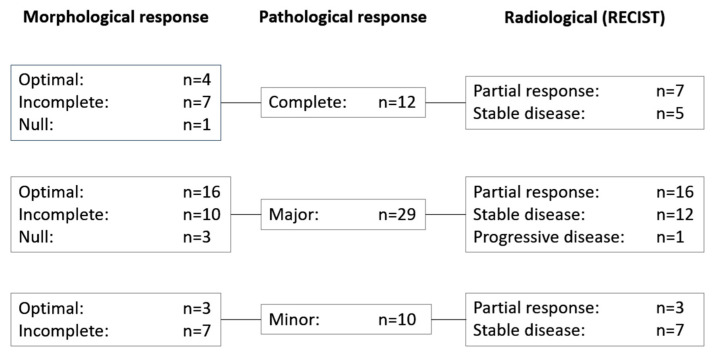
Correlation of response data in patients with morphological, pathological, and radiological response data (*n =* 51). RECIST, response evaluation criteria in solid tumors.

**Figure 3 cancers-12-02259-f003:**
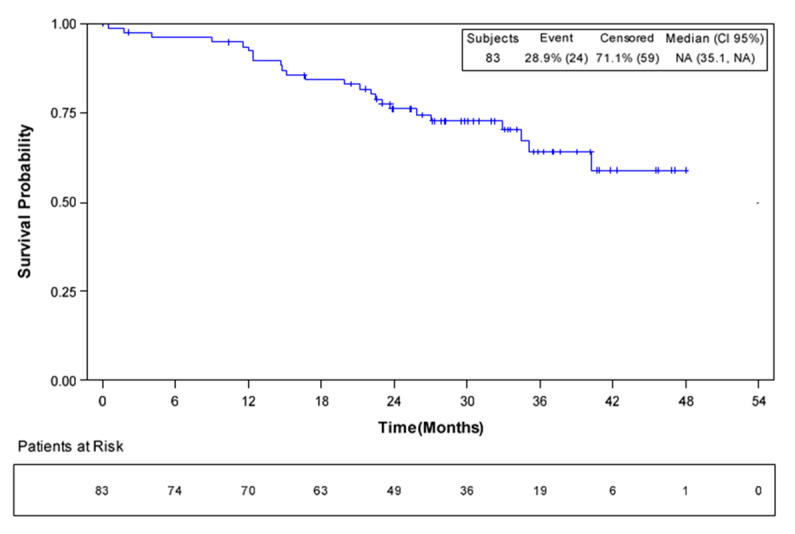
Overall survival in all patients (*n* = 83).

**Table 1 cancers-12-02259-t001:** Patient characteristics at baseline (*n* = 83).

Characteristic	All (*n =* 83)	*KRAS* Mutant (*n =* 36) ^a^	*KRAS* Wild Type (*n =* 43)
Median age, years (IQR)	66 (58–71)	67 (57–72)	66 (58–71)
Sex, *n* (%)			
Male	55 (66)	23 (64)	29 (67)
Female	28 (34)	13 (36)	14 (33)
ECOG performance status, *n* (%)			
0	64 (77)	27 (75)	34 (79)
1	18 (22)	9 (25)	9 (21)
Missing	1 (1)	0	0
Tumor location, *n* (%)			
Colon	46 (55)	21 (58)	24 (56)
Rectum	26 (31)	10 (28)	15 (35)
Both	2 (2)	1 (3)	1 (2)
Other	9 (11)	4 (11)	3 (7)
Histological grade, *n* (%)			
Grade 1	17 (20)	8 (22)	8 (19)
Grade 2	45 (54)	20 (56)	24 (56)
Grade 3	8 (10)	6 (17)	2 (5)
Grade 4	2 (2)	0	2 (5)
Grade X	11 (13)	2 (6)	7 (16)
Disease stage, *n* (%)			
Not classified	7 (8)	2 (6)	4 (9)
I	1 (1)	1 (3)	0
IIA	10 (12)	6 (17)	4 (9)
IIB	1 (1)	1 (3)	0
III	3 (4)	2 (6)	1 (2)
IIIA	1 (1)	1 (3)	0
IIIB	2 (2)	1 (3)	1 (2)
IV	58 (70)	22 (61)	33 (77)
Prior systemic chemotherapy, *n* (%)	13 (16)	8 (22)	5 (12)

IQR, interquartile range; ^a^
*KRAS* mutation status was not available for four patients.

**Table 2 cancers-12-02259-t002:** Response rates in evaluable population and according to *KRAS* mutation status.

Response Category, *n* (%)	Overall	*KRAS* Mutant	*KRAS* Wild Type
Radiological response ^a^	*(n = 66)*	*(n = 31)*	*(n = 35)*
Complete	0	0	0
Partial	32 (48)	16 (52)	16 (45)
Stable disease	29 (44)	12 (39)	17 (49)
Progressive disease	5 (7)	3 (9)	2 (9)
Morphological response	*(N = 65)*	*(N = 30)*	*(N = 35)*
Optimal	26 (40)	14 (47)	12 (34)
Incomplete	33 (50)	14 (47)	19 (54)
Null	6 (9)	2 (6)	4 (11)
Pathological response	*(N = 51)*	*(N = 22)*	*(N = 29)*
Complete	12 (24)	3 (14)	9 (31)
Major	29 (57)	14 (64)	15 (52)
Minor (no response)	10 (19)	5 (22)	5 (17)

^a^ Response evaluation criteria in solid tumors version 1.1.

**Table 3 cancers-12-02259-t003:** Adverse events occurring in >5% of patients (safety population; *n* = 78).

Adverse Event, *n* (%)	All Grades	Grade 1/2	Grade 3/4
Diarrhoea	51 (65)	41 (53)	10 (13)
Asthenia	45 (58)	40 (51)	5 (6)
Nausea	31 (40)	31 (40)	0
Neurotoxicity	28 (36)	22 (28)	6 (8)
Paresthesia	26 (33)	26 (33)	0
Vomiting	24 (31)	21 (27)	3 (4)
Reduced appetite	23 (29)	22 (28)	1 (1)
Peripheral neuropathy	13 (17)	12 (15)	1 (1)
Mucosal inflammation	13 (17)	12 (15)	1 (1)
Abdominal pain	12 (15)	11 (14)	1 (1)
Palmar-plantar erythrodysesthesia	11 (14)	8 (10)	3 (4)
Neutropenia	10 (13)	8 (10)	2 (3)
Anemia	9 (12)	8 (10)	1 (1)
Epistaxis	5 (6)	4 (5)	1 (1)
Hypertension	5 (6)	4 (5)	1 (1)
Rectal hemorrhage	4 (5)	4 (5)	0
Proteinuria	4 (5)	4 (5)	0
Pulmonary embolism	4 (5)	1 (1)	3 (4)

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
