# Peer review of "Correlation of RECIST, Computed Tomography Morphological Response, and Pathological Regression in Hepatic Metastasis Secondary to Colorectal Cancer: The AVAMET Study"

_cancers, 2020, doi:10.3390/cancers12082259_

Round 1
Reviewer 1 Report
It is a major problem with the methodology of the AVAMET study that it cannot answer the clinically relevant question wether or not the CT morphological criteria (CTMC) improves assessment of resectability of colorectal liver metastases in patients treated with bevacizumab. A control group of patients NOT receiving bevacizumab (and blinded for the radiologists) is needed. Another problem is that it is unclear if the CTMC attributes were considered or not in the preoperative evaluation of resectability.
Correlations were calculated (and were primary endpoint) of pathological response in resected CRC liver metastases with RECIST-defined response and with CTMC in patients with resectable metastases receiving neoadjuvant chemotherapy with capecitabine, oxaliplatin and bevacizumab. Resectability was initially based on a baseline CT scan and criteriae were <4 mets, and all < 10 cm i.e. a matter of number and size as in the RECIST classification. It is not clearly described but must be anticipated that the same criteriae were pivotal for whether surgery was undertaken or nor after 4 series of chemotherapy? The crucial clinical question in that situation is if addition of CTMC further qualifies or introduces doubt in the surgeons decision to operate the patient or not. Correlations were poor between RECIST and CTMC and between CTMC and pathological response observed in the resected metastases (kappa tests resulted in poor agreement) suggesting that CTMC characterization is a waste of time. Fourteen patients did not have surgery, four because of progression (RECIST?) and 10 because of chemo toxicity, but how could chemo toxicity prevent surgery?
Having the current material the trialists can be recommended to let a new panel of radiologists CTMC score the CT scans to put figures on the degree of reproducibility and, secondly, to form a panel of radiologists and surgeons and let them go through all cases who were alive at he time when surgery was scheduled at inclusion in the trial, asking for an assessment of resectability based on the scans and the RECIST attributes followed by addition of CTMC attributes and compare outcomes with original assessments and to which extent CTMC added anything to their decision making.
Tabel on bevacizumab toxicity data is far out of the scope of the paper.
Author Response
Reviewer-1
It is a major problem with the methodology of the AVAMET study that it cannot answer the clinically relevant question wether or not the CT morphological criteria (CTMC) improves assessment of resectability of colorectal liver metastases in patients treated with bevacizumab. A control group of patients NOT receiving bevacizumab (and blinded for the radiologists) is needed. Another problem is that it is unclear if the CTMC attributes were considered or not in the preoperative evaluation of resectability.
Correlations were calculated (and were primary endpoint) of pathological response in resected CRC liver metastases with RECIST-defined response and with CTMC in patients with resectable metastases receiving neoadjuvant chemotherapy with capecitabine, oxaliplatin and bevacizumab. Resectability was initially based on a baseline CT scan and criteriae were <4 mets, and all < 10 cm i.e. a matter of number and size as in the RECIST classification. It is not clearly described but must be anticipated that the same criteriae were pivotal for whether surgery was undertaken or nor after 4 series of chemotherapy? The crucial clinical question in that situation is if addition of CTMC further qualifies or introduces doubt in the surgeons decision to operate the patient or not. Correlations were poor between RECIST and CTMC and between CTMC and pathological response observed in the resected metastases (kappa tests resulted in poor agreement) suggesting that CTMC characterization is a waste of time. Fourteen patients did not have surgery, four because of progression (RECIST?) and 10 because of chemo toxicity, but how could chemo toxicity prevent surgery?
Having the current material the trialists can be recommended to let a new panel of radiologists CTMC score the CT scans to put figures on the degree of reproducibility and, secondly, to form a panel of radiologists and surgeons and let them go through all cases who were alive at he time when surgery was scheduled at inclusion in the trial, asking for an assessment of resectability based on the scans and the RECIST attributes followed by addition of CTMC attributes and compare outcomes with original assessments and to which extent CTMC added anything to their decision making.
Tabel on bevacizumab toxicity data is far out of the scope of the paper
Responses to reviewer -1
We appreciate your suggestions. You are right that AVAMET trial cannot answer the question whether or not the CTMC improves assessment of resectability but the endpoint of our trial was only correlations between CTMC, RECIST criteria and pathological response. In fact, metastases must be considered initially resectable to include the patient in the trial. In our mind, if correlation between CTMC and pathological regression had been positive, radiological response on CTMC could be used as prognostic factor. Finally, though morphological response (optimal/incomplete) was more sensitive than RECIST (complete/partial response) for predicting pathological responses (complete/major), Kappa coefficients indicated a lack of correlation between the results of RECIST and morphological responses and between morphological and pathological response rates.
Furthermore, you are right regarding resectability criteria were the same initially and after chemotherapy.
You are right noting that fourteen patients did not have surgery because of progression or toxicity. But this number only includes patients who followed the study protocol, undergoing resection after receiving treatment according to the trial design. Some patients with toxicity underwent surgery but were not included in the analysis of correlations. We pointed it in the second paragraph of discussion (lines 305 to 308).
We will take into account your suggestions regarding reproducibility for future trials on the same topic.
Reviewer 2 Report
This is a well written manuscript which would be of interest to many readers. There are a few minor typos in the manuscript that should be addressed. There are also some discrepancies in Table 2 which should also be addressed.
I would encourage the authors to expand on the discussion regarding the relationship between CTMR and histopathological findings. Is there any association between solid/ cystic nature of the tumour? tumour grade? tumour heterogeneity?
Minor typos noted:
Line 177 (referencing number should be superscript)
Line 206 (space missing) "51 of"
Line 216 -Table 2 -
The numbers reported in this table don't add up between the overall cohort and the KRAS mutant and KRAS wild type cohorts across all the radiological responses, morphological responses. If cases are missing from certain sub-cohorts this should be explained in the legend or included in the table.
Line 326 (referencing number should be superscript)
Author Response
Reviewer-2
This is a well written manuscript which would be of interest to many readers. There are a few minor typos in the manuscript that should be addressed. There are also some discrepancies in Table 2 which should also be addressed.
I would encourage the authors to expand on the discussion regarding the relationship between CTMR and histopathological findings. Is there any association between solid/ cystic nature of the tumour? tumour grade? tumour heterogeneity?
Minor typos noted:
Line 177 (referencing number should be superscript)
Line 206 (space missing) "51 of"
Line 216 -Table 2 -
The numbers reported in this table don't add up between the overall cohort and the KRAS mutant and KRAS wild type cohorts across all the radiological responses, morphological responses. If cases are missing from certain sub-cohorts this should be explained in the legend or included in the table.
Line 326 (referencing number should be superscript
Responses to reviewer -2
We appreciate your suggestions. Thank you for your comments.
CTMC are based in three issues: heterogeneity and attenuation, tumour–liver interface, and peripheral rim enhancement. Unfortunately, they have not been independently related to histological findings. It is possible that one of these issues were more sensitive to predict histological regression than CTMC grades. We include this suggestion in the discussion:
… CTMC are based in three issues: heterogeneity and attenuation, tumour–liver interface, and peripheral rim enhancement. It is possible that one of these issues, in an independent way, were more sensitive to predict histological regression….
Minor typos have been corrected:
Line 177 – Done
Line 206 – Done
Line 216 – Table 2 has been corrected
Line 326 – Done